# First Evidence of CpGV Resistance of Codling Moth in the USA

**DOI:** 10.3390/insects13060533

**Published:** 2022-06-10

**Authors:** Jiangbin Fan, Johannes A. Jehle, Ann Rucker, Anne L. Nielsen

**Affiliations:** 1Key Laboratory of National Forestry and Grassland Administration on Management of Forest Bio-Disaster, College of Forestry, Northwest A&F University, Xianyang 712100, China; 2Department of Entomology, Rutgers, The State University of New Jersey, Bridgeton, NJ 08302, USA; rucker@njaes.rutgers.edu; 3Institute for Biological Control, Julius Kühn Institute (JKI)–Federal Research Centre for Cultivated Plants, 69221 Dossenheim, Germany; johannes.jehle@julius-kuehn.de

**Keywords:** resistance testing, discriminating concentration, insect, Tortricidae, baculovirus, rearing, field application

## Abstract

**Simple Summary:**

Different isolates of Cydia pomonella granulovirus (CpGV) have been formulated into different biological control agents that are applied to reduce the infestation and losses induced by codling moth in organic and integrated apple and pear orchards. Cyd-X^®^ is the most widely applied CpGV product in apple orchards in the USA and contains a single active ingredient, the isolate CpGV-M (GV-0001). Aiming to investigate the susceptibility of codling moth to GV-0001, we used a discriminating virus concentration to screen five codling moth populations derived from Washington State (USA), which were reared in the laboratory (colony WA1-WA5). The results indicate that colony WA3 had a significantly reduced susceptibility to GV-0001 and survivors from the bioassay can produce offspring. WA3 represents the first documented case of CpGV resistance in the USA. In further tests, WA3 showed resistance to all commercial CpGV agents currently registered in the USA. Nonetheless, the resistant WA3 colony showed high susceptibility to three newly developed CpGV preparations. Therefore, the introduction of the novel CpGV formulations may allow the maintenance of sustainable management programs for codling moth in the USA.

**Abstract:**

Codling moth (*Cydia pomonella* L.) is a very important pest in apple, pear, and walnut orchards worldwide, including the USA. Cydia pomonella granulovirus (CpGV) is used to control codling moth in organic and conventional production. Due to increasing codling moth infestations from organic apple orchards in Washington State, USA, five codling moth colonies (WA1-WA5) were screened for their susceptibility relative to the isolate GV-0001, the main active ingredient of Cyd-X^®^, using a discriminating concentration of 6 × 10^4^ OB/mL. Compared to a susceptible laboratory colony, the observed results indicated that GV-0001 lacked efficacy against codling moth colony WA3. It was confirmed that WA3 was the first case of codling moth resistance to CpGV in the USA. Further testing of WA3 was performed on a range of CpGV isolates and a lack of efficacy was observed against additional isolates. However, three newly developed CpGV preparations can efficiently infect larvae from the resistant colony WA3. Our results suggest that there is an urgent need to monitor the situation in the USA, aiming to prevent the emergence or spread of additional codling moth populations with CpGV resistance. Strategies to sustain the efficacy of codling moth control using novel CpGV formulations need to be developed.

## 1. Introduction

Codling moth (*Cydia pomonella* L.) (Lepidoptera: Tortricidae) is a temperate insect species, and during the larval stage, it can damage pome fruit (apple and pear), stone fruit (apricot, plum), and walnut, etc., through internal feeding. Codling moth invaded North America over 200 years ago and has since become a very serious pest in apple, pear, and walnut production [1]. Codling moth larvae bore deep into the core of unripe and ripening apples, called the “worm in the apple”, resulting in direct yield loss when it is not well controlled under the economic threshold [1,2]. In the USA, there are two to three annual generations of codling moth [3]. Overwintering diapausing larvae pupate in early spring and emerging moths mate and begin to lay eggs right after fruit trees bloom, typically during petal fall, at which time the first instars will bore into the developing fruit. During feeding, frass pushed out through the entrance hole (or “sting”) becomes an obvious sign of an infested fruit [4,5]. The larvae leave the fruit at the fifth instar to pupate under the bark of tree trunk or other protected locations in the orchard. The successive generations continue feeding on developing fruit, although depending on harvest times, infested fruit is not always readily observable and may result in contaminated fruit. Although chemical pesticides can control codling moth, there are non-target risks and documented resistance to pyrethroids and other chemicals [6,7,8]. In addition, the consumers’ demand for organic fruit directly pushes pest control strategies towards biological methods. For example, mating disruption using phermones, spinosad, and commercial formulations of Cydia pomonella granulovirus (CpGV) have been applied to reduce codling moth population in organic and conventional orchards [9,10,11,12].

CpGV is a double-strand DNA virus belonging to the genus *Betabaculovirus* of the family *Baculoviridae* [13]. The occlusion body (OB) of CpGV with an average size of 360 × 210 nm (length × width) contains a single virion [14]. CpGV is very specific and highly virulent to codling moth larvae. The median lethal dose (LD_50_) is 1.37~28 OBs for first instars (L1) and 10~92 OBs for fifth instars (L5) [15], depending on the experimental setup. CpGV was developed as a biological control agent in Europe and North America in the 1970s and 1980s and was first registered in the late 1980s [16]. Viral OBs are ingested by codling moth larva, resulting in the initial infection of the midgut from where infection spreads to other tissues. At the end of infection, the larval body is full of newly produced Obs and the larvae turn whitish or are milky in color. CpGV-infected codling moth larvae will succumb within 3–7 days post-infection but delayed lethality occurs in some populations, taking up to 14 days until the larvae die. When CpGV products are applied for control of codling moth, the efficacy rates of 75% to 90% can be achieved in the field [17].

CpGV agents have been developed into the most broadly applied biological control products for managing codling moth in organic and integrated pest management (IPM) orchards around the world, due to its host specific, highly effective, and environmentally sound characteristics [9,18,19]. Since CpGV was first isolated from an orchard in Mexico, termed CpGV-M [20], more than forty CpGV isolates have been collected and grouped into seven phylogenetic lineages, namely genome group A to G [20,21]. Isolates from genome groups A, E, and B, such as CpGV-M, CpGV-S, and CpGV-E2, have been developed into different commercial formulations (Table 1). CpGV was further isolated, selected, and passaged on oriental fruit moth, *Grapholita molesta* Busck (Lepidoptera: Tortricidae), for many generations with high virulence against both codling moth and oriental fruit moth.

After many years of successful application of commercial products that were based on CpGV-M, codling moth populations in southern Germany and France were identified to be 1000- to 100,000-fold less susceptible to CpGV [22,23]. These findings were the first documented cases of host resistance to commercial baculovirus biocontrol products. The resistance of codling moth field populations to products containing CpGV-M was termed type I and was further documented in other European countries, e.g., the Netherlands, Switzerland, Austria, Italy, and Czech Republic, threatening local organic apple production [23,24,25]. Meanwhile, extensive resistance testing and genetic studies revealed three types (I to III) of field resistance to CpGV with distinguishable patterns of inheritance and differing susceptibility to CpGV isolates belonging to different genome groups [23,26]. Beyond that, newly discovered types of resistance have occurred in France and Italy [27]. The most common type I resistance has been characterized as dominant and Z chromosomal inherited, highly stable, and probably with no fitness costs [23,28]. Laboratory-selected and naturally collected CpGV isolates can overcome current resistance types [23,26,29,30].

Apple is one of the most valuable pome fruit varieties in the USA and more than 90% of USA organic apple are produced in Washington State [31]. In these orchards, CpGV products are an important component of the pest management program and are often paired with mating disruption. Several CpGV products have been used for decades on 8100–12,200 ha in North America [9]. Currently, CpGV products of Cyd-X^®^, Cyd-X HP^®^, Madex HP^®^ and ViroSoft CP4^®^ are being used in the USA. The frequent application of CpGV per generation for successive years and an increase in infested fruit leads to the question of CpGV resistance in the USA.

Some organic apple orchards in Washington State noted high codling moth infestation despite frequent CpGV application. We collected overwintering larvae from five of these orchards in Washington State to test their susceptibility to CpGV products. We applied a protocol using a discriminating CpGV concentration to differentiate between susceptible and resistant codling moth larvae [23]. We further aimed to determine the resistance status of these colonies against different CpGV products and formulations and provided possible solutions to overcome CpGV resistance using newly developed CpGV products.

## 2. Materials and Methods

### 2.1. Insects and Viruses

A susceptible laboratory colony (LabS) was derived from Benzon Research (Carlisle, PA, USA) insect rearing company. This colony was established more than 20 years ago at the Yakima Agricultural Research Laboratory in Washington State. Five populations (WA1-WA5) collected as diapausing larvae from organic apple orchards in eastern Washington State in the winter of 2018–2019 were used. Diapausing larvae were kept under dark conditions at 10 °C until pupation. Adults and progeny larvae were reared in a climate chamber at 25 °C under the photoperiod of 16/8 h (light/dark) at Rutgers Agricultural Research and Extension Center in Bridgeton, NJ, USA. All codling moth colonies were maintained on an artificial diet that is composed of corn meal, wheat germ, beer yeast, agar, ascorbic acid, two preservatives of Nipagin and Benzoic acid, and water [32].

Virus stocks of different commercial products and test formulations (Table 1) were stored at −20 °C. The OB concentration was counted using a Petroff-Hausser chamber (depth 0.02 mm) under the dark field of a light microscope (Nikon ECLIPSE E600).

### 2.2. Activity of CpGV Formulations on Codling Moth Larvae

For resistance testing, virus OBs were suspended in molecular grade water to a concentration of 3 × 10^6^ OB/mL, and a total volume of 1 mL was incorporated into 49 mL of semi-artificial diet [32] once it had cooled down to 45 °C, resulting in a final concentration of 6 × 10^4^ OB/mL derived from the LC_95_ value of the susceptible codling moth colony against CpGV-M in laboratory bioassays [23,28]. At this so-called discriminating concentration, the mortality of fully susceptible neonate larvae (L1) is expected to be >95% after 7 days post infection (dpi) and up to 100% after 14 dpi, whereas the average mortality of CpGV-resistant codling moth (CpR) is significantly decreased [23]. Therefore, this discriminating concentration was applied to monitor the susceptibility of L1 larvae of LabS and WA1 to WA5. After mixing the diet with OBs, 50 mL of diet was poured into an autoclavable 50-well tray (LICEFA, Bad Salzuflen, Germany) and allowed to dry at room temperature for 1 day. The dimension of each well is 1.5 × 1.5 × 1.5 cm (length × width × height). One freshly hatched (0–12 h old) codling moth larva (L1) was transferred to each well. The same procedure was performed for the untreated control, with 1 mL of molecular grade water incorporated into the diet. Larvae of the LabS colony and WA1-WA5 were all evaluated using the same protocol. Thirty to fifty larvae were evaluated for each replicate in resistance test, and three replicates of each isolate were evaluated if not otherwise indicated. Any colony indicating less than 30% mortality to GV-0001 (Cyd-X^®^) at 7 dpi was further bioassayed against seven other CpGV isolates/products (Table 1) at the discriminating OB concentration [23]. Dead individuals were recorded at 1, 7, 14, and 21 dpi. Larvae that died on day 1 because of handling were excluded from the experiment. If needed, the mean mortality at 7, 14, and 21 dpi was corrected for control mortality according to Abbott’s formula [33]. Data were analyzed with *t*-test and one-way ANOVA followed by multiple comparisons using Tukey–Kramer HSD for significant difference analysis in JMP PRO 15. The results were plotted in GraphPad Prism 9.1 (GraphPad Software, San Diego, CA, USA).

## 3. Results and Discussion

### 3.1. Resistance Test of WA Colonies

When the larvae of the LabS colony and five WA colonies were exposed to GV-0001 at the discriminating concentration of 6 × 10^4^ OB/mL, the mortalities of LabS (n = 3 replicates, N = 146 larvae), WA1 (n = 1, N = 36), WA2 (n = 1, N = 29), WA3 (n = 3, N = 143), WA4 (n = 1, N = 35), and WA5 (n = 2, N = 65) were 95.4%, 100.0%, 100.0%, 15.9%, 41.8%, and 100% at 7 dpi, respectively (Figure 1). Compared to the larvicidal activity of GV-0001 on the internal reference LabS, colonies WA3 and WA4 showed a reduced susceptibility. On the other hand, WA3 had 41.8% and 67.3% mortality at 14 and 21 dpi, respectively, and the WA4 colony showed a mortality of 67.7% and 100% at 14 and 21 dpi, respectively. Taken together, the resistance test showed that WA3 was less susceptible than the other colonies and is considered to be partially resistant to GV-0001 (Cyd-X^®^), which is composed of CpGV-M (genome group A) (Table 1). It is likely that also WA4 shows partial resistance to GV-001 because mortality at day 7 and day 14 was also considerably reduced when compared to LabS or WA1, WA2, and WA5, which showed typical >95% mortality of a susceptible colony when challenged with the discriminating concentration. However, the number of reared larvae was not sufficient for performing additional replicates for WA1, WA2, WA4, and WA5 (Figure 1). Statistical analyses were only conducted for LabS and WA3 colonies, showing that the mortalities of LabS and WA3 induced by GV-0001 were significantly different at 7, 14, and 21 dpi, respectively (*p* < 0.05) (Figure 1).

According to product specifications, CpGV products should be applied every 7–8 sunny days, which may easily exceed 10 sprays during a single apple growing season in some regions in Washington State. Comparing the CpGV application history in Europe and the current result, a lack of efficacy of GV-0001 has occurred in Washington State. CpGV-exposed larvae from the WA3 colony have successfully pupated and emerged into adults (data not shown), indicating that resistance to GV-0001 can be passed to the next generation. Laboratory resistance testing using artificial diet mixed with CpGV OBs can achieve reliable results to recognize CpGV-resistant codling moth populations. It is, therefore, a good indicator if a given codling moth population is resistant to CpGV agents and products [34,35].

### 3.2. Activity Test of CpGV Formulations to WA3

After identifying that GV-0001 had reduced efficacy on the WA3 colony, we tested seven other available CpGV preparations containing different CpGV isolates (Figure 1). All CpGV preparations showed a high virulence to LabS colony (Appendix A), while the WA3 colony had a significantly different susceptibility against CpGV preparations in the tests at 7 dpi (F = 7.64, df = 7, *p* < 0.05), at 14 dpi (F = 7.51, df = 7, *p* < 0.05), and at 21 dpi (F = 6.02, df = 7, *p* < 0.05) (Figure 2). At 7 dpi, the mortality caused by GV-0001, GV-0014 (Madex HP^®^), and CpGV-S (Virosoft CP4^®^) was significantly lower than that induced by GV-0015, GV-0013, and GV-0017, respectively (*p* < 0.05). At 14 dpi, the mortality caused by GV-0014 and CpGV-S was significantly lower than that induced by GV-0015, GV-0013, and GV-0017, respectively (*p* < 0.05). At 21 dpi, only CpGV-S induced mortality was significantly lower than that induced by GV-0015, GV-0013, and GV-0017, respectively (*p* < 0.05). The WA3 colony demonstrated low susceptibility to CpGV isolates of the genome groups A and E but not relative to genome group B. This pattern of susceptibility is similar to type II or type III resistance [26]. All newly developed isolates of GV-0015, GV-0013, and GV-0017 consisting of or containing a portion of CpGV belonging to genome group B appear to be more virulent to WA3 larvae than the other CpGV preparation in all tests (Figure 2). These findings are akin to previous reports of resistance tests, and CpGV isolates belonging to genome group B were able to overcome all current types of resistance [21,26]. After 21 dpi, the surviving larvae completed development into adults. When the adults were single-pair crossed, they laid viable eggs, which were used to establish a genetic homogeneous WA3 colony (data not shown), indicating that gene responses to CpGV-resistance were present in the wild WA3 population. Similar characteristics of field codling moth populations containing resistant and susceptible individuals have been observed in European countries [26,36].

Here, for the first time in the USA, we document the resistance of codling moth to CpGV-M (= GV-0001) in Washington State, USA, indicating that the genetic background to establish CpGV resistance is not restricted to codling moth populations in Europe. To further investigate the resistance observed in the WA3 colony, the mechanism and inheritance of resistance require further investigation. As a first step, inbreeding and further selection is necessary to establish a genetically homogenous line. Single-pair crossing experiments and resistance tests will then be needed to identify the mode of inheritance and to evaluate if the noted CpGV resistance of WA3 is indeed similar to type II or type III resistance [23,26]. From our assays, it can be concluded that it does not follow the resistance pattern of type I resistance given the lack of susceptibility to CpGV variants belonging to genome group A or E (GV-0001, GV-0003, GV-0014, GV-0006, and CpGV-S) [23,26]. Learning from the lessons on CpGV resistance in Europe, it is important to carefully monitor the performance of currently used CpGV products in the USA and to conduct more resistance testing of codling moth populations where high infestations and control failures of CpGV products are noted [22,25,27]. On the other hand, CpGV application in the USA and elsewhere needs to be carefully and properly carried out in accordance to the appropriate degree-day timing and life stage target and weather conditions [9,19,37], aiming to use CpGV products in the most sustainable way possible. With the identification of CpGV preparations causing sufficient mortality of WA3, novel CpGV products could be made available to growers to overcome high infestations rates caused by codling moth.

## 4. Conclusions

The results indicated that GV-0001, the main isolate of Cyd-X^®^, lacked efficacy against WA3 colony and a third of inoculated WA3 larvae survived until the end of the bioassay. One other colony, WA4, has become less susceptible to GV-0001, suggesting that resistance may be more widespread than indicated by our study. It was further confirmed that colony WA3 was the first reported case of CpGV resistance in the USA and WA3 shows resistance to all commercial CpGV products available on the US market. Most importantly, three newly developed CpGV formulations showed similar efficacy as Cyd-X^®^, in susceptible codling moth populations offering new possibilities to improve control efficacy and successfully kill CpGV-resistant CM populations in the USA.

## Figures and Tables

**Figure 1 insects-13-00533-f001:**
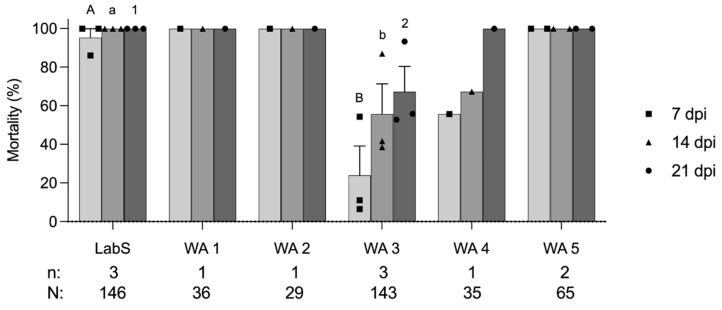
Mortality (mean ± standard error) at 7, 14, and 21 days post-infection (dpi) of six codling moth colonies LabS, WA1 to WA5 exposed to 6 × 10^4^ OB/mL GV-0001 (Cyd-X^®^). Each data point representing mortality at 7, 14, and 21 dpi was plotted as a square, triangle, and circle, respectively. Data were analyzed with *t*-test at *p <* 0.05. Different capital letters, lowercase letters, and numbers represent the significant differences of mortality at 7, 14, and 21 dpi, respectively. The number of replicates (n) and the total number of tested individuals (N) of each codling moth colony are shown below the chart.

**Figure 2 insects-13-00533-f002:**
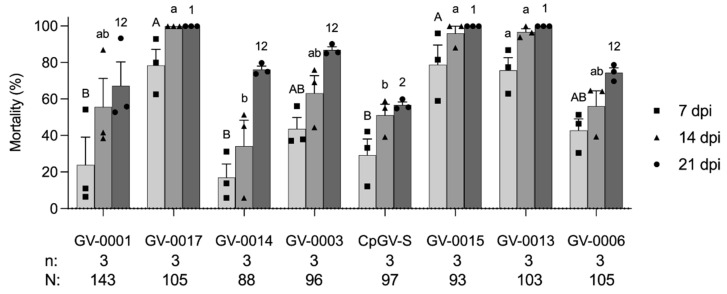
Mortality (mean ± standard error) of first instars of codling moth from WA3 colony exposed to eight CpGV formulations at a concentration of 6 × 10^4^ OBs/mL. Mortality was recorded at 7, 14, and 21 days post infection (dpi). Each data point representing the mortality at 7, 14, and 21 dpi was plotted as a square, triangle, and circle, respectively. Data were analyzed with one-way ANOVA followed by Tukey–Kramer HSD comparison at *p <* 0.05. Different capital letters, lowercase letters, and numbers represent the significant differences of mortality at 7, 14, and 21 dpi, respectively. All tested individuals (N) and replicates (n) are shown below the chart.

**Table 1 insects-13-00533-t001:** Commercial and experimental CpGV products/formulations in the USA and Europe. Preparations used in this study are marked with an asterisk *.

Isolates	Genome Group ^1^	USA (Product/Formulation)	Europe (Product/Formulation)
GV-0001	A	Cyd-X^®^ *	Madex^®^
GV-0003	A	Cyd-X HP^®^ *	Madex Plus^®^
GV-0014	A	Madex HP^®^ *	Madex Twin^®^
GV-0006	A + E	-	Madex Max^®^ *
GV-0015	B	-	Madex Primo^®^ *
GV-0013	B + E	-	Madex Top^®^ *
GV-0017	A + B + E	Madex XLV^®^ **	Madex Duo^®^ *
CpGV-S	E	ViroSoft CP4^®^ *	-

^1^ Summary of genome group of CpGV products/formulations is derived from [21]. ** Madex XLV is under review for an experiment use permit (EUP) in the USA as CX-6485.

## Data Availability

The data presented in this study are available on request from the corresponding author.

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
