# Peer review of "First Evidence of CpGV Resistance of Codling Moth in the USA"

_insects, 2022, doi:10.3390/insects13060533_

Round 1

Reviewer 1 Report

In this work the authors test the efficacy of different CpGV strains on Cydia pomonela in strains originating from the US. At least in one population isolate CpGV-M demonstrates low efficacy and I think that indeed this is the first record of reduced activity of a widespread CpGV product in the US and in this respect, Ι consider the work and the results as valuable. Moreover, the objectives and the methodology of the study are clearly demonstrated and the results are well presented.

Some general comments that could be addressed by the authors are the following:

  1. You should describe in more detail how you have derived the final dose used for the bioassays. I think just taking a commercial dose per se is not enough, considering that the application in lab conditions does not always reflect the field application. For example, traditionally, in lab bioassays, different ''doses'' are used to detect first the LC50 and LC90 efficacy of the component. Such an approach also provides the confidence intervals of a potential dose efficacy.
  2. Which were the criteria used to classify the population as different strains. Do they not come from the same area? i.e. How far are the different populations located from each other?
  3. Moreover, why and how was the C. pomonella susceptible strain was judged as a susceptible one in first place?
  4. You should comment more on the potential differentiation (or not) in evaluating the efficacy of the CpGV when mixed on diet compared to the standard application of CpGV formulations.
  5. Is there any case that the preservatives included in the artificial diet affects the survival rate of the larvae? Please explain. Furter, in this respect, maybe you should consider to listing in details the components of the diet that you have used for the bioassays.
  6. What could be the reason that one particular population shows lower susceptibility over the other? Again, aren’t they from the same region? Is there any physical barrier or other reason to justify these differences in CpGV susceptibility? Somthing elsε?

Author Response

Response to Reviewer´s comments:

We appreciate the reviewer’s comments on our manuscript of “First evidence of CpGV resistance of codling moth in the USA” which are very valuable to improve the quality of the manuscript. We have carefully revised the manuscript according to the reviewer´s suggestions with the aim to present the data in a more clearly and easily understandable way, as well as using a more active voice to highlight our results and findings. Attached is the point-to-point response to the reviewer' comments.

Reviewer 1

In this work the authors test the efficacy of different CpGV strains on Cydia pomonela in strains originating from the US. At least in one population isolate CpGV-M demonstrates low efficacy and I think that indeed this is the first record of reduced activity of a widespread CpGV product in the US and in this respect, Ι consider the work and the results as valuable. Moreover, the objectives and the methodology of the study are clearly demonstrated and the results are well presented.

Some general comments that could be addressed by the authors are the following:

  1. You should describe in more detail how you have derived the final dose used for the bioassays. I think just taking a commercial dose per seis not enough, considering that the application in lab conditions does not always reflect the field application. For example, traditionally, in lab bioassays, different ''doses'' are used to detect first the LC50 and LC90 efficacy of the component. Such an approach also provides the confidence intervals of a potential dose efficacy.

Response: In previous studies it was found that resistant codling moth populations can exhibit a 1,000 to 100,000-fold decrease of LC50 values. In these cases it is not possible to determine an LC50 or LC90 as only a small cohort of tested animals will die in a bioassay or extremely high amounts of virus will be needed in an assay. Therefore the concept of a discriminating concentration was developed. The concept is based on a LC95 of susceptible codling moth larvae as outlined by Asser-Kaiser et al. 2006 and many other publications. If a susceptible L1 larvae of codling moth is exposed to a concentration of 6 × 104 OB/ml, it is expected that >95% of larvae will succumb within 7 days.  Schmitt et al., (2013) described that “if 30% of a population is resistant, the concentration/mortality curve reaches a plateau at 70% mortality. Even with high CpGV-M doses, no increase in mortality will occur further. This in consequence influences the slope of the concentration/response curve. The more data points at high concentration are included in the calculation of the concentration/response curve, the flatter the curve will be, and correspondingly, LC50 values will increase. Therefore, the layout of bioassay biases the values of concentration/response curves.” In order to avoid the bias of increasing LC50 values in laboratory bioassays, the discriminating concentration of 6 × 104 OB/ml that was originally obtained from the ~95% lethal concentration (LC95) of the susceptible codling moth strain (CpS) against CpGV in the 7-day was therefore applied to measure the susceptibility of codling moth field populations composed of susceptible and resistant individuals in this study.

Since 2006, resistance screening using the discriminating concentration of 6 × 104 OB/ml has been developed into a fast and reliable approach to identify codling moth resistance to CpGV in Germany and other European countries, as well as to determine and reflect the resistance ratio of codling moth field populations in several projects.

The detailed explanation has been added in the part of Materials and Methods.

  1. Which were the criteria used to classify the population as different strains. Do they not come from the same area? i.e. How far are the different populations located from each other?

Response: The original text was somehow confusing about the use of the terms “population”, “strain” and “colony”. We had collected diapausing larvae from 5 orchards with a distance between 15.6 to 139 km. Since we had no information about the genetic relationship to each other, each of these samples were considered to represent a wild population of codling moth. After taking them to the laboratory and continued rearing there, we considered each of them as a “colony”. We further avoided to use the term “strain”. A “strain” would result from further selection and would refer to a genetically homogenous collection of individuals obtained from mass crossing or single crossing experiments. As this was not the case, we omitted the term “strain” in the revision. The text of the revision was adapted accordingly.

  1. Moreover, why and how was the C. pomonellasusceptible strain was judged as a susceptible one in first place?

Response: We consider a codling moth strain or colony as fully susceptible if a seven day exposure to the discriminating concentration of 6 × 104 OB/ml of CpGV-M results in a >90-95% mortality in our test system. This mortality corresponds to a LC50 of about 1-3 × 103 OB/ml. There a numerous studies, which were cited in the manuscript, where this approach was used and described. We have included the description of the criteria of a susceptible strain in the text. 

  1. You should comment more on the potential differentiation (or not) in evaluating the efficacy of the CpGV when mixed on diet compared to the standard application of CpGV formulations.

Response: Comments concerning the efficacy of CpGV formulations have been added in the results and discussion. There is no potential differentiation in evaluating the efficacy of the CpGV between laboratory bioassays (fed on diet) and standard application (spraying in orchard) (Eberle et al., 2008; Zingg et al., 2011). Of course, we do not really know the exposure of a codling moth larvae to CpGV in the field but this is not necessary to evaluate the standard procedure of resistance testing. Based on the results of more than 120 field collected codling moth population the resistance test using the discriminating concentration proved to be a highly valuable tool to distinguish between susceptible and resistant codling moth. If a population was identified as resistant it always corresponded with high damage observed in the field despite regular CpGV sprays. 

Moreover, the resistance-breaking CpGV isolates can induce the same mortality on the resistant codling moth, whenever it is conducted in field trial and laboratory bioassay, indicating no differentiation is observed in two approaches of bioassays.

Zingg, D.; Züger, M.; Bollhalder, F.; Andermatt, M., Use of resistance overcoming CpGV isolates and CpGV resistance situation of the codling moth in Europe seven years after the first discovery of resistance to CpGV-M. IOBC-WPRS Bull 2011, 66, 401-404.

Eberle, K. E.; Asser-Kaiser, S.; Sayed, S. M.; Nguyen, H. T.; Jehle, J. A., Overcoming the resistance of codling moth against conventional Cydia pomonella granulovirus (CpGV-M) by a new isolate CpGV-I12. J Invertebr Pathol 2008, 98, 293-298.

  1. Is there any case that the preservatives included in the artificial diet affects the survival rate of the larvae? Please explain. Furter, in this respect, maybe you should consider to listing in details the components of the diet that you have used for the bioassays.

Response: The artificial diet we adopted in the bioassays has been used to rear the codling moth for 50 years of research. No side effect  on CpGV activity or larval health was ever observed or reported which were related the two preservatives of Nipagin and Benzoic acid added in the artificial diet. The components of artificial diet have been listed in this manuscript.

  1. What could be the reason that one particular population shows lower susceptibility over the other? Again, aren’t they from the same region? Is there any physical barrier or other reason to justify these differences in CpGV susceptibility? Somthing elsε?

Response: Five codling moth field populations were collected from five different locations separated by Yakama river and Columbia river, as well as several mountains. The natural landscape hinders their spreading activity. The WA3 colony was sampled in a small valley and was far away from other orchards, which hardly allow natural exchange with other populations.  We only could speculate about the reasons. But experience from Europe shows that CpGV application history in orchards and genetic background may play a major role. Because natural exchange of CM between orchards is low, it is hypothesized that CpGV resistance is independently selected in different orchards. We do not consider insect behavior but human activity, e.g. a transportation of apple boxes, as the main factor for codling moth spread. 

Reviewer 2 Report

This manuscript provides an evaluation of resistance of codling moth populations to different strains of the entomopathogen Cydia pomonella granulovirus (CpGV). Unfortunately, I cannot recommend this manuscript in the present form. Results description does not match with data plotted in figure 1; also, such data must be statistically analysed; lettering in figure 2 is odd; discussions must be strongly improved. I have the following comments:

-L15: definition of CpGV should be clarified also here.

-L68-69: are these ranges? If yes, more details must be provided as the ranges seems very variable.

-L104-112: the aim of experimentation and research hypotheses must be better introduced.

-L135: more details must be added in this section. For instance, what is the dimension of each well?

-L137: please be more specific. Was it one larva to each well?

-L147-148: the software used for the analyses must be specified.

-L156: according to Fig. 1, only WA3 and WA4 are lower than the experimental control.

-L157: mortality of WA4 at 14 dpi is not 100% according to fig. 1.

-Figure 1: please provide statistical analysis of these data.

 -L177, L179: these are results of anova and not of multiple comparison procedure. Please, specify that such results are reported in the figure.

-L182-206: this part should be the discussion section. Unfortunately, it is extremely short and poorly organized (only two papers have been cited!). I suggest to extensively reword this section and include discussion of more published papers.

-Figure 2: I suggest to completely revise the results of multiple comparisons. For instance, WA3 mortality exposed to GV-0017, GV-0015 and GV-0013 seem similar, but the numbering is different.

-correct typos in the reference list, (e.g., ref 15).

Author Response

Response to Reviewer´s comments:

We appreciate the reviewer’s comments on our manuscript of “First evidence of CpGV resistance of codling moth in the USA” which are very valuable to improve the quality of the manuscript. We have carefully revised the manuscript according to the reviewer´s suggestions with the aim to present the data in a more clearly and easily understandable way, as well as using a more active voice to highlight our results and findings. Attached is the point-to-point response to the reviewer' comments.

Review 2

This manuscript provides an evaluation of resistance of codling moth populations to different strains of the entomopathogen Cydia pomonella granulovirus (CpGV). Unfortunately, I cannot recommend this manuscript in the present form. Results description does not match with data plotted in figure 1; also, such data must be statistically analysed; lettering in figure 2 is odd; discussions must be strongly improved. I have the following comments:

  1. -L15: definition of CpGV should be clarified also here.

Response: It has been revised.

  1. -L68-69: are these ranges? If yes, more details must be provided as the ranges seems very variable.

Response: The cited publication [15] refers to a review where different studies were compared. The ranges came from experiments carried out in different laboratories over several decades. The differences were most likely caused by different experimental set-ups, different experimental duration, and likely even by different quality of OB counting.

  1. -L104-112: the aim of experimentation and research hypotheses must be better introduced.

Response: They have been clarified in introduction.

  1. -L135: more details must be added in this section. For instance, what is the dimension of each well?

Response: more detail has been added into this section. The dimension of each well is 1.5 × 1.5 × 1.5 cm (length: width: height).

  1. -L137: please be more specific. Was it one larva to each well?

Response: It has been revised in the part of materials and methods. Yes, one larva was placed to each well of the tray.

  1. -L147-148: the software used for the analyses must be specified.

Response: It has been revised. Statistical software of JMP PRO 15 was used in this study.

  1. -L156: according to Fig. 1, only WA3 and WA4 are lower than the experimental control.

Response: It has been revised and reworded.

  1. -L157: mortality of WA4 at 14 dpi is not 100% according to fig. 1.

Response: It has been revised. The mortality of WA4 at 14 dpi is 67.7%.

  1. -Figure 1: please provide statistical analysis of these data.

Response: Statistical analyses was performed for LabS and WA3. There were not enough additional replicates for the other populations. Reporting their mortality appears to be important as they reached high mortality which indicates high susceptibility in these populations.

  1. -L177, L179: these are results of anova and not of multiple comparison procedure. Please, specify that such results are reported in the figure.

Response: It has been revised and specified.

  1. -L182-206: this part should be the discussion section. Unfortunately, it is extremely short and poorly organized (only two papers have been cited!). I suggest to extensively reword this section and include discussion of more published papers.

Response: This section has been reworded. Several papers have been added in the discussion and compared with the finds of this study.

  1. -Figure 2: I suggest to completely revise the results of multiple comparisons. For instance, WA3 mortality exposed to GV-0017, GV-0015 and GV-0013 seem similar, but the numbering is different.

Response: The results of multiple comparisons were analyzed with one-way ANOVA, followed by multiple comparison using Tukey-Kramer HSD for significant difference analysis in JMP PRO 15.

  1. -correct typos in the reference list, (e.g., ref 15).

Response: It has been corrected.

Reviewer 3 Report

This manuscript screening the CpGV-resistant codling moth in USA. This manuscript is well written, but the length and scientific finding of this manuscript are more like communication. There are only several points that need to be clarified.

1)    In the summary, the author mentioned “Nonetheless, three newly developed CpGV formulations can efficiently control the resistant population. Therefore, introduction of the novel CpGV formulations may allow maintenance of sustainable management programs for codling moth in the USA.” But it did not mention in the abstract. 
2)    Since three newly developed CpGV formulations can efficiently control the resistant population. Did the author mean that controlling WA3?
3)    In figure 1, WA3 showed the variation, please present each data point if there are replicates. 
4)    Please indicate each data point in figure 2

Author Response

Response to Reviewer´s comments:

We appreciate the reviewer’s comments on our manuscript of “First evidence of CpGV resistance of codling moth in the USA” which are very valuable to improve the quality of the manuscript. We have carefully revised the manuscript according to the reviewer´s suggestions with the aim to present the data in a more clearly and easily understandable way, as well as using a more active voice to highlight our results and findings. Attached is the point-to-point response to the reviewer' comments.

Review 3

This manuscript screening the CpGV-resistant codling moth in USA. This manuscript is well written, but the length and scientific finding of this manuscript are more like communication. There are only several points that need to be clarified.

1)    In the summary, the author mentioned “Nonetheless, three newly developed CpGV formulations can efficiently control the resistant population. Therefore, introduction of the novel CpGV formulations may allow maintenance of sustainable management programs for codling moth in the USA.” But it did not mention in the abstract. 

Response: This information has been added in the abstract.

2)    Since three newly developed CpGV formulations can efficiently control the resistant population. Did the author mean that controlling WA3?

Response: yes, three newly developed CpGV formulations of GV-0013, GV-0015, GV-0017 can efficiently control WA3 resistant population. As shown in Figure 2, the mortality of WA3 induced by GV-0013, GV-0015, GV-0017 were 75.8%, 78.9% and 78.5% in 7-day resistance test. And the mortality reached to 100% in 21-day resistance test. No offspring can be produced.

3)    In figure 1, WA3 showed the variation, please present each data point if there are replicates. 

Response: each data point has been added in Figure 1.

4)    Please indicate each data point in figure 2

Response: each data point has been added in Figure2.

Round 2

Reviewer 2 Report

In this revised version Authors have only addressed minor changes. I was suggesting to conduct statistical analyses to data in Figure 1 but this has not been done. Although they have changed a bit the text (L169-185) no reasonable justification was provided for the missing analysis, nor did they reply to my comment.

Figures 1 and 2 are poorly prepared as the data points at 14 and 21 dpi are non visible in the pictures 

L70 and elsewhere: use the proper symbol "×" as in L69 and not "x"

L134: "presentatives"?

L153: replace ":" with "×"

L154: abbreviate "hours" with "h"

References: the references must be completely revised. e.g., scientific names in refs. 24, 25, 26 must be italicized

Author Response

It is to express our gratitude to the reviewer. You carefully went through our manuscript and proposed many impressive questions and comments that help us to improve the manuscript a lot. Attached are the point-to-point respond to the reviewer(s)' comments.

  1. In this revised version Authors have only addressed minor changes. I was suggesting to conduct statistical analyses to data in Figure 1 but this has not been done. Although they have changed a bit the text (L169-185) no reasonable justification was provided for the missing analysis, nor did they reply to my comment.

Response: The statistical analyses in general requires at least three replates to calculate and compare the sample mean and the sample variance. Because there were not enough additional replicates for WA1, WA2, WA4 and WA5 colonies, the statistical analyses cannot be performed using one-way ANOVA or t-test for all colonies. There was a single replicate for the colonies of WA1, WA2 and WA4, were two replicates for WA5 colony and were three replicates for LabS and WA3 (Figure 1). Statistical analyses were therefore performed for LabS and WA3 using t-test at p < 0.05.

  1. Figures 1 and 2 are poorly prepared as the data points at 14 and 21 dpi are non visible in the pictures 

Response: It has been revised.

  1. L70 and elsewhere: use the proper symbol "×" as in L69 and not "x"

Response: It has been revised.

  1. L134: "presentatives"?

Response: It has been revised into “preservatives”

  1. L153: replace ":" with "×"

Response: It has been revised.

  1. L154: abbreviate "hours" with "h"

Response: It has been revised.

  1. References: the references must be completely revised. e.g., scientific names in refs. 24, 25, 26 must be italicized

Response: The references have been carefully revised. It is written in the guidelines of International Committee on Taxonomy of Viruses (ICTV) that the full virus species name shall be written in italic, but the name of virus isolate shall be written in regular. The Refs 24, 25, 26 presented the isolate name of virus, thereby scientific names in refs. 24, 24 26 should be regular but not be italicized.

Reviewer 3 Report

The authors have addressed my suggestions. It could be accpted  by the journal.

Author Response

Thank you very much for your suggestions for style required and language editing in this manuscript. We have carefully revised it and address data in a more clearly way.